# The Signaling Pathway of the ADP Receptor P2Y_12_ in the Immune System: Recent Discoveries and New Challenges

**DOI:** 10.3390/ijms24076709

**Published:** 2023-04-04

**Authors:** Philomena Entsie, Ying Kang, Emmanuel Boadi Amoafo, Torsten Schöneberg, Elisabetta Liverani

**Affiliations:** 1Department of Pharmaceutical Sciences, School of Pharmacy, College of Health Professions, North Dakota State University, Fargo, ND 58105, USA; 2Division of Molecular Biochemistry, Rudolf Schönheimer Institute of Biochemistry, Medical Faculty, Leipzig University, 04103 Leipzig, Germany

**Keywords:** P2Y_12_ signaling pathway, immune system, antiplatelet therapy

## Abstract

P2Y_12_ is a G-protein-coupled receptor that is activated upon ADP binding. Considering its well-established role in platelet activation, blocking P2Y_12_ has been used as a therapeutic strategy for antiplatelet aggregation in cardiovascular disease patients. However, receptor studies have shown that P2Y_12_ is functionally expressed not only in platelets and the microglia but also in other cells of the immune system, such as in monocytes, dendritic cells, and T lymphocytes. As a result, studies were carried out investigating whether therapies targeting P2Y_12_ could also ameliorate inflammatory conditions, such as sepsis, rheumatoid arthritis, neuroinflammation, cancer, COVID-19, atherosclerosis, and diabetes-associated inflammation in animal models and human subjects. This review reports what is known about the expression of P2Y_12_ in the cells of the immune system and the effect of P2Y_12_ activation and/or inhibition in inflammatory conditions. Lastly, we will discuss the major problems and challenges in studying this receptor and provide insights on how they can be overcome.

## 1. Introduction

P2Y_12_ is a G-protein-coupled receptor (GPCR) found first on platelets and microglia and is activated by ADP but also by ATP binding [1]. Activation of the P2Y_12_-mediated signaling pathway leads to platelet aggregation and potentiation of degranulation [2] as well as migration of microglia [3]. Recent studies successfully proved that P2Y_12_ is expressed in a wider selection of cells, especially in immune cells [4,5,6,7,8]. Indeed, P2Y_12_ mRNA was detected in monocytes [6], dendritic cells [4], macrophages [9], megakaryocytes [10], and T lymphocytes [8]. Additional studies have shown that P2Y_12_ is also functionally expressed at the protein level in immune cells. These observations raised the question of whether blocking P2Y_12_ could be beneficial for the outcome of inflammatory diseases in animal models and ultimately in humans. Several studies were carried out, showing that blocking P2Y_12_ can be beneficial for the outcome of various inflammatory conditions, such as sepsis [11,12], asthma [13,14], rheumatoid arthritis [14,15,16], and neuroinflammation [17,18]. However, there is still a disparity of results between studies, probably due to the numerous challenges in investigating the signaling pathways involved. In this review, we will summarize our current knowledge about P2Y_12_ expression and function in the immune system and the effects of blocking P2Y_12_ in animal models of diseases and patients. We will discuss the challenges encountered in these studies and lastly, we will provide suggestions and insights on how they could be overcome.

## 2. The ADP Receptor P2Y_12_

The ADP receptor P2Y_12_ is a GPCR that couples primarily to G_i_ proteins [19]. In platelets, activation of P2Y_12_ potentiates agonist-induced dense granule release, pro-coagulant activity, and thrombus formation [20]. Specifically, α-granule release and subsequent surface expression of p-selectin [21] have been noted. In platelets, adenylyl cyclases generate cyclic adenosine monophosphate (cAMP) in response to prostacyclin (prostaglandin I_2_, PGI_2_) and Prostaglandin E_1_, which are prostanoids secreted by healthy endothelial cells. However, P2Y_12_ activation causes inhibition of adenylyl cyclase activity, leading to a decrease in intracellular cAMP levels [22] and also activation of phosphoinositide 3-(PI3) kinase (Figure 1). P2Y_12_ activation may support Ca^2+^ mobilization by activation of the phospholipase β2 by the G-protein subunits βγ [23], and ADP also activates other ADP receptors in platelets, e.g., P2Y_1_.

ADP binding to platelets produces selective short-term (5–10 min) desensitization of P2Y_12_ resulting in unresponsiveness to the subsequent addition of agonists [24]. P2Y_12_–mediated desensitization is mediated by GPCR kinases (GRK) 2 and 6 [25].

Previous studies have shown that G_i_ signaling mediated by P2Y_12_ is dependent on cholesterol-rich lipid rafts [26] and a high-fat diet enhances platelet activation induced by other agonists [27]. In addition to chronic hypercholesterolemia, other pathologic conditions ranging from diabetes [28] to hypertension may increase P2Y_12_ receptor functions and hence the risk of thrombosis.

Furthermore, this receptor is essential for platelet aggregation under shear conditions as P2Y_12_ inhibition was able to decrease shear-induced platelet aggregation [29], causing a diminished p-selectin expression and microparticle formation initiated by the von Willebrand factor (vWF) activation [30]. However, greater inhibition was observed when P2Y_12_ was antagonized [31]. Similar results were observed in a mouse model of atherothrombosis, where pre-treatment with the P2Y_12_ antagonists ticagrelor or cangrelor could inhibit thrombus formation and decrease its stability [32]. Similar results were observed in ex vivo thrombus formation with human platelets from coronary heart disease patients treated with clopidogrel [33].

Defects in the gene encoding the P2Y_12_ receptor are responsible for a congenital bleeding disorder [34]. Patients with defective P2Y_12_ receptor functions have normal platelet shape change but impaired abilities to inhibit adenylyl cyclase activity [35]. Dense granules are normal in both numbers and content, but granule release is generally decreased.

## 3. Expression of the ADP Receptor P2Y_12_ in the Immune System

Considering the promising data for P2Y_12_ antagonists as a treatment for improving inflammatory diseases [4,11,13,36,37], there has been increasing interest in investigating specifically whether the effects of P2Y_12_ inhibition were exerted exclusively on platelets or cells of the immune system. So far, studies have been performed in monocytes/macrophages [6], T lymphocytes [7,8], dendritic cells [4], and neutrophils [5] (Table 1). To date, no studies have investigated P2Y_12_ function in B lymphocytes and Natural Killer cells.

### 3.1. Platelets

The ADP receptor P2Y_12_ has been reported to play a role in the aggregation of platelets [20]. P2Y_12_ has been reported as the most successful in targeting platelets due to its key role in thrombosis. There is platelet adhesion and ADP release from dense granules upon platelet exposure to collagen and vWF [21]. The coupling of platelet P2Y_12_ to G_i_ proteins prevents adenylyl cyclases’ cAMP levels from decreasing, which may contribute to an increase in the activation state of the platelets. Therefore, ADP- and collagen exposition of human platelets lacking P2Y_12_ impairs aggregation and secretion [40]. Platelets lack a nucleus but still contain mRNA which may be attributed via translation to platelet functions [38]. It is yet to be established whether P2Y_12_ expression occurs during platelet maturation and/or prior to their release by megakaryocytes [39].

### 3.2. Monocytes and Macrophages

It is still under debate whether monocytes express P2Y_12_ mRNA. In one study, physiological and pharmacological modulation of the two ADP receptors’ (P2Y_1_ and P2Y_12_) crosstalk could influence Ca^2+^ signaling in monocytes [6]. A database search using the keywords monocytes and purinergic signaling revealed P2Y_12_ expression in human CD14^+^/CD16^−^ monocytes (https://www.ebi.ac.uk/gxa/home, accessed on 29 March 2023). On the other hand, P2Y_12_ mRNA was detected more consistently in macrophages. Single-cell RNA sequencing revealed high expression of P2Y_12_ in macrophages isolated from various tissues, e.g., lung, skin, prostate, and breast (https://gtexportal.org/home/gene/P2RY12#singleCell, accessed on 29 March 2023). Moreover, P2Y_12_ expression was measured in RNA isolated from resting human monocyte-derived macrophages [43,44]. P2Y_12_ expression was confirmed on CD68^+^ CD163^+^ tumor-associated macrophages of melanoma in situ where P2Y_12_ triggers the migration of macrophages towards nucleotide-rich, necrotic tumor areas, and modulates the inflammatory environment upon ADP binding [45]. These data were confirmed by another group [43]: differentiated macrophages express P2Y_12,_ and they migrate towards ADP. They show that macrophage migratory functions can be directly inhibited by P2Y_12_ receptor antagonists, reflecting direct anti-inflammatory properties. This is in contrast with another study where P2Y_12_ receptor ligands are not chemotactic for macrophages [46], and P2Y_12_ receptor antagonists act indirectly in a platelet-dependent manner on monocytes as anti-inflammatory agents. Despite macrophage activation appearing to be regulated by PGI_2_ and, therefore, by changes in cAMP intracellular levels [51], no experiments have fully explored whether the effect of blocking P2Y_12_ signaling pathways is due to alterations in cAMP levels. Taken together, these data suggest that P2Y_12_ can be expressed in both monocytes and macrophages, but more studies are required to show both mRNA and protein levels, especially in primary human macrophages. However, the functional relevance of P2Y_12_ in monocytes is still unclear.

### 3.3. T Lymphocytes

One of the first studies on T lymphocytes has shown that ADP-induced CD45^+^ leukocyte migration was significantly reduced in P2Y_12_-null mice as compared to the wild-type (WT) controls [50]. However, a more recent study did show that P2Y_12_ deficiency did not influence cell differentiation and proliferation of CD4^+^ T cells in vitro [18] while the authors report a higher level of T helper 17 (Th17) cells in P2Y_12_-knockout (KO) mice, in the animal model of multiple sclerosis, experimental autoimmune encephalomyelitis (EAE) [18]. Further studies in EAE confirmed that P2Y_12_ deficiency alleviates EAE symptoms by reducing the Th17 differentiation [48]. However, in this paper, the authors show that P2Y_12_ can directly regulate Th17 differentiation in vitro [48]. This is in line with our previous work in an animal model of sepsis where P2Y_12_ antagonism alters regulatory T cell (Treg) population size and function in vivo and in vitro [11]. These data overall suggest that P2Y_12_ activation regulates T-cell differentiation.

We have also investigated whether P2Y_12_ is a potential target for ADP in T cells. Our results show that ADP exposure changes T-cell proliferation and cytokine secretion in a timely- and stimulus-specific manner, indicating that P2Y_12_ expressed by T lymphocytes is functional [7]. T-lymphocyte activation appears to be regulated by PGI_2_ and cAMP intracellular levels [49]. Therefore, changing cAMP intracellular levels could be the mechanism through which P2Y_12_ blocking could alter T-cell response. Interestingly, we have investigated changes in cAMP levels in peripheral blood mononuclear cells (PBMCs) upon ADP exposure and P2Y_12_ blockade, and despite cAMP levels being altered, it appears to be P2Y_12_-independent [7]. These data indicate that inhibiting P2Y_12_ function may target not only platelets but also T-lymphocyte activation. Notably, ADP also exerts P2Y_12_-independent effects on T lymphocytes that may be due to the stimulation of other purinergic receptors.

### 3.4. Neutrophils and Eosinophils

We have investigated whether neutrophils express P2Y_12_ by treating them with a prasugrel metabolite mixture generated in vitro [5]. This mixture was able to inhibit neutrophil functions, most likely indirectly, although platelet P2Y_12_ inhibition did not abolish prasugrel metabolite effects, suggesting the possible off-target effects of this drug [5]. However, it has been reported that eosinophils express P2Y_12_ at a protein level [13,47]. Blocking P2Y_12_ appeared to alter eosinophil infiltration in the fibrotic liver [47]. Hence, clopidogrel’s effect in decreasing asthma [13] and bacterial infection [47] could be due to its effects on this cell type.

### 3.5. Dendritic Cells

Dendritic cells (DCs) appear to express P2Y_12_ mRNA, and these cells are functionally altered in P2Y_12_-null mice [4]. These cells also express P2Y_13_ mRNA, but DCs isolated from P2Y_13_-null mice did not show different functions as compared with the cells isolated from the WT mice. The authors show that P2Y_12_ activation stimulates endocytosis and antigen-presenting functions [4]. Moreover, blocking P2Y_12_ diminished ADP-induced Ca^2+^ mobilization. In another study with bone marrow-derived dendritic cells (BMDCs), loss of P2Y_12_ significantly increased the production of IL-23 in contrast to WT BMDCs [18]. Interestingly, P2Y_12_-deficient DCs promoted more naïve CD4^+^ T cells to differentiate into Th17 cells. P2Y_12_ receptors can affect the cytokine profile of the BMDCs [18]. As activation of DCs appears to be sensitive to PGI_2_ exposure and, therefore, by changes in cAMP intracellular levels [49], it would be interesting to investigate whether the effects of blocking P2Y_12_ signaling pathways are due to alteration in cAMP levels. Overall, it appears that modulating P2Y_12_ signaling pathways in DCs alters their functions, but further studies are required to determine the mechanism.

### 3.6. Microglia

Microglial cells are the resident immune cells in the brain, and hence they are essential for tissue maintenance as well as immune responses during neuroinflammation [3]. They physically interact with other glial cells and neurons, and they assess the environment to ensure tissue health [52]. P2Y_12_ is expressed in most microglia subpopulations (Figure 2A) and its expression appears to be age-related and sex-dependent [53,54]. Microglia chemotaxis, which is essential for tissue repair and tissue clearance during inflammation [55], appears to be dependent on P2Y_12_ in humans and mice [3]. Furthermore, P2Y_12_ signaling seems to be important also for the microglia-neurons crosstalk [56]. Previous studies, summarized in the review by Illes et al. (2020), indicate that purinergic receptor antagonists could alter microglia activation state [57]. P2Y_12_ has been also investigated at a protein level and it appeared to be also altered when the microglia are activated as compared with quiescent microglia [57]. However, when considering these data, it is important to address the challenges in studying GPCRs at the protein level (see the section of the paper on “Challenges in studying the receptor P2Y_12_”). In Figure 2B, changes in GPCR mRNA levels in homeostatic or disease-associated microglia (DAM) are shown. Interestingly, P2Y_12_ mRNA decreases when microglia are activated as compared with the homeostatic counterpart.

Several signaling pathways downstream to P2Y_12_ appear to be involved in microglial chemotaxis. For instance, recent data suggest that microglial P2Y_12_ could be involved in the nod-like receptor protein 3 (NLRP3) inflammasome activation [62]. The aberrant activation of this inflammasome signaling has been demonstrated to contribute to the development of several neurological diseases [63]. Another study suggested that activation of P2Y_12_ in the central nervous system regulates microglial activation via the RhoA/ROCK pathway [64]. This could be the mechanism through which activation of P2Y_12_ contributes to inflammation in the microglia.

### 3.7. Other Cells Relevant to Inflammation

P2Y_12_ has also been investigated in smooth muscle cells (SMC) and endothelial cells. Binding of ADP to the receptor P2Y_12_ decreases cAMP and increases intracellular calcium levels, resulting in cell contractions. These cells are not part of the immune system but still play an essential role in the cardiovascular system during inflammatory processes. In SMC, both mRNA and protein expression were detected [50,65]. Furthermore, Wilhborg et al. showed that AR-C67085, a P2Y_12_ antagonist, was able to prevent ADP-induced contractions [65], suggesting that this receptor blockage could alter SMC functions. Another group investigated 2-MeS-ADP-mediated SMC migration in P2Y_12_-null mice, where migration was significantly inhibited in null mouse cells compared with WT. Although interesting, these data should be supported by further analysis of intracellular changes in cAMP levels or Akt phosphorylation to demonstrate receptor functionality.

P2Y_12_ expression has also been detected in both human and rat endothelial cells. For example, a study has detected P2Y_12_ expression in rat splenic sinus endothelial cells [66] by Western blotting and fluorescence microscopy. However, no studies have demonstrated whether the receptor is functional in vivo, and hence we do not know whether blocking P2Y_12_ could alter endothelial cell functions in the splenic artery. Moreover, no study has yet investigated whether P2Y_12_ is expressed in other vessels as well. Human pulmonary microvascular endothelial cells seem to express P2Y_12_ mRNA and receptor protein and it is increased upon treatment with LPS [67]. Blocking P2Y_12_ with ticagrelor or clopidogrel seemed to protect the cells from inflammation, by reducing LPS-induced cytokine mRNA levels, promoting cell migration, and overall improving cell functions. Previous data have indicated that the protective effects of P2Y_12_ blockage were through modulation of the immune response, while these new findings suggest a wider role of P2Y_12_. In addition, it would also be interesting to exclude that the effects noted are not P2Y_12_-independent by repeating the experiments in P2Y_12_ knockout pulmonary endothelial cells.

## 4. Drugs Targeting P2Y_12_

Considering P2Y_12_’s importance in the platelet function [20], thienopyridines, a class of P2Y_12_-specific antagonists, have been designed and successfully used to prevent thrombus formation [68]. These drugs bind to P2Y_12_ and therefore prevent ADP-induced aggregation of platelets and consequent thrombus formation [20,69]. The most common drugs in this class are clopidogrel, prasugrel, and ticagrelor [70]. These drugs inhibit platelet aggregation by modulating adenylate cyclase activity and, therefore, regulate cAMP levels inside platelets. They have been generally well-tolerated in patients, but there are side effects, such as the risk of bleeding, that still need to be taken into consideration [71]. There are recommended doses provided to clinicians [71], but overall practice guides have not been fully provided. As a result, it is not uncommon for physicians to switch drugs and adapt the therapy based on patients.

Clopidogrel has been used worldwide, and it is a very well-established drug [72]. There are over two decades of experience using this drug [73]. It is administered orally as a prodrug and metabolized into its active form by a two-step enzymatic process in the liver [73]. This process is catalyzed by the cytochrome P450 (CYP450) enzyme which is encoded by the CYP2C19 gene [73]. Polymorphisms of this gene can lead to changes in the kinetics of clopidogrel metabolism resulting in inter-individual variation in therapeutic drug levels. There are several limitations to the drug: slow onset of action (3–8 h), moderate level of inhibition of platelet aggregation, and high variability within the population [70]. P2Y_12_-independent effects have been reported for clopidogrel, suggesting that this drug should be further evaluated [74].

Prasugrel is also a prodrug, administered orally, and rapidly hydrolyzed by esterases in the intestine and blood to a thiolactone intermediate metabolite (R-95913) [75]. This intermediate metabolite undergoes subsequent activation by a single CYP450–dependent step (predominantly CYP3A and CYP2B6) to form the sulfhydryl-containing active metabolite (R-138727) [75]. Prasugrel has a more rapid action, and it is more potent than clopidogrel [76]. Prasugrel has shown less variability between patients, and therefore it can be used as an alternative to clopidogrel non-responders. It is usually administered at a lower dose. Prasugrel has shown higher bleeding risks than other antiplatelet drugs [77] but this has not been noted in all the studies [78]. This discrepancy needs to be taken into consideration when prescribed. However, the risk of bleeding appears to increase with age, so prasugrel may not be the best drug for the elderly [79].

Ticagrelor has a different chemical structure compared to clopidogrel and prasugrel [80] and does not require metabolic activation following oral administration [70]. Moreover, the binding of ticagrelor to P2Y_12_ is reversible with a faster offset of platelet inhibition than clopidogrel [81]. Ticagrelor has shown higher bleeding [78] than clopidogrel but not consistently in all studies [82]. Ticagrelor has also the disadvantage of a more frequent dose application and higher cost [70]. Pleiotropic effects have been noted when ticagrelor was administered. For example, ticagrelor was able to inhibit NLRP3 inflammasome activation in mouse macrophages and in peripheral mononuclear cells from patients with acute coronary syndrome, independently of P2Y_12_ [83]. Further investigations are required to determine whether these P2Y_12_-independent effects provide unexpected advantages to the immune system and, therefore, can be employed as inflammatory drugs. On the other hand, it could also cause negative effects, especially when administered in combination with other drugs [84].

Cangrelor (AR-C 69931) is an intravenous, reversible, short-acting P2Y_12_ blocker in vitro and ex vivo [5,7,20,85,86,87]. It is not a prodrug, so it does not require metabolic activation [88]. In several studies, [88,89,90,91] it did not show a significant difference from clopidogrel or ticagrelor [92].

These drugs have been investigated not only to prevent thrombus formation but also to influence inflammation in a variety of animal models, such as LPS-induced inflammation [93], cecal ligation and double puncture [12], myocardial infarction [94], and pancreatitis [94,95]. It is still unclear whether the effect of thienopyridines on inflammation is mediated by influencing platelet functions or if it is caused by direct modulation of immune cells expressing P2Y_12_.

## 5. Challenges in Studying the Receptor P2Y_12_

Studying P2Y_12_ has presented certain challenges that have most likely caused the discrepancy in data collected so far, such as the lack of a specific antibody or impurity of the cell population analyzed. We would like to premise that it is overall challenging to detect GPCRs at the protein level. As membrane proteins, they are low-expressed and glycosylated [96]. Moreover, they tend to form SDS-resistant multimers [97] which makes immune detection in Western blots difficult. On the other hand, not detecting a GPCR in a tissue or a cell using immunological methods does not necessarily imply that the receptor is not expressed. Therefore, it is advisable to perform control experiments using KO mouse material. We now discuss the major challenges in this chapter (Table 2).

### 5.1. Reliable Antibody

Other P2Y receptors are widely expressed in immune cells [8] and P2Y receptors have even areas of similar primary structures [98]. As a result, the major challenge has been to find a reliable antibody that specifically recognizes the protein P2Y_12_ over other P2Y receptors (e.g., P2Y_13_). Analyzing mRNA content can overcome this challenge, as shown in numerous experiments [4,6,8] but mRNA content is not always a direct proxy of functional P2Y_12_. Radio-ligand-receptor binding studies with P2Y_12_ are not an easy task because nucleotide tracers, such as [^3^H]2MeSADP, have multiple binding sites in cells and tissues, and specific and high-affinity isotope-labeled antagonists are rare. Another way to overcome this challenge is to measure the effects of P2Y_12_ activation in the absence and presence of receptor antagonists. This method can provide valid information about functionality, but a reliable technique for determining the protein expression of P2Y_12_ is still needed.

### 5.2. Antagonist Specificity

Ligand specificity is an important issue in P2Y_12_ research because there are evolutionarily and pharmacologically related nucleotide receptors. For example, P2Y_13_, another member of the P2Y receptor family, is a G_i_-coupled receptor activated by ADP which has a similar pharmacological profile to P2Y_12_ [99]. AR-C67085 is specific for P2Y_12_ at low concentrations (nM), but at higher concentrations (μM) it can also bind to P2Y_13_ [100,101]. Hence, using the appropriate concentration appears to be crucial to obtain reliable data. Applying various antagonists at the same time can also be useful for comparing different P2Y results. For example, we investigated P2Y_13_ receptor blockade with MRS2211 alongside blocking P2Y_12_ in T-lymphocyte functions [7]. Our data show that blocking P2Y_12_ or P2Y_13_ alters T cells differently, providing information specifically for these different signaling pathways [7].

In vivo, we used different drugs to block P2Y_12_, and this has contributed to creating discrepancies in the outcome. For instance, prasugrel and clopidogrel are administered as pro-drugs and they need to be metabolized [72]. This generates discrepancies as different individuals may metabolize the drug differently. Moreover, several metabolites besides the active thiol derivate have been described for clopidogrel [74]. These different metabolites may also explain the P2Y_12_-independent effects of P2Y_12_ antagonists [5,12]. Ticagrelor should reduce the problem as it is not a prodrug because it does not need to be metabolized for antiplatelet action. However, another study has shown that ticagrelor can inhibit the LPS-induced activation of the NLRP3 inflammasome in macrophages independently of P2Y_12_ [83].

### 5.3. Cell Purity

When analyzing single-cell type preparation from a primary source, there is always the chance of contamination. Indeed, some studies were performed on cell lines with clearly defined purity. Data with cells isolated from mice or humans have higher relevance and validity, but impurity may mislead interpretations. One of the problems is due to the small size of platelets, as they can easily be trapped in other cell layers, or they may be overlooked when certain methods are employed for cell counting. A small number of platelets could then contaminate the population of immune cells and lead to false positive detection of P2Y_12_ mRNA or protein content.

Moreover, during inflammation, activated platelets express p-selectin on their surface and, therefore, can adhere to immune cells such as monocytes and T lymphocytes. This contributes to the challenge of isolating immune cells without being contaminated by platelets and their aggregates. Hence, selecting an accurate method to isolate primary cells could help to reduce such contaminations.

## 6. P2Y_12_ Activation in the Immune System during Inflammatory Conditions

Although cell lines and primary cells can provide certain insights into P2Y_12_ function, these data need to be verified in animal models or/and human patients. The activation of P2Y_12_ in the immune system has been studied in both animal models and human patients. The following chapter and Table 3 summarize the most recent findings in different disease states such as infections, sepsis, asthma, arthritis, and atherosclerosis.

### 6.1. Atherosclerosis

Atherosclerosis is the narrowing and hardening of an artery caused by plaque formation originating within the endothelial lining [105]. Atherosclerosis has been indicated as a chronic inflammatory disease characterized by a chronic, low-grade inflammatory response that recruits cells of innate and adaptive immunity into the atherosclerotic plaque [106]. As mentioned above, P2Y_12_ has been found in different cell types of the vascular, such as endothelial cells [67], vascular smooth muscle cells [65], and immune cells [4,5,6,7] which suggests a role of P2Y_12_ in cardiovascular physiology and as a pharmacological target in related diseases. Indeed, some clinical studies have shown that P2Y_12_ blockers are more effective than aspirin in patients with ischemic stroke/transient ischemic attack (TIA) of atherosclerotic origin [107,109]. The pharmacologic blockade [110,111] or genetic deletion [112] of P2Y_12_ can improve atherosclerosis lesion development, evidenced by a reduction in plaque size, an increase in fiber content of the plaques, and a decrease in inflammatory molecule levels (e.g., P-/E-selectin) [110,111]. P2Y_12_ expression on the platelet surface is increased by nicotine or high glucose. Platelets are then easily activated, causing damage to endothelial cells and increased levels of secreted inflammatory molecules [113]. These inflammatory molecules recruit inflammatory cells, such as monocytes and neutrophils, followed by the occurrence of inflammatory cascades [113]. In addition, the SMC P2Y_12_ promotes the proliferation and migration of SMCs into the plaque [114,115]. Thus, P2Y_12_ is a very promising target for treating atherosclerosis, and P2Y_12_ receptor blockers may be a good option for secondary prevention of atherosclerotic ischemic stroke, which needs to be further investigated [116].

### 6.2. Rheumatoid Arthritis

Rheumatoid arthritis (RA) is an autoimmune disorder characterized by progressive synovial inflammation and swelling. Although a disease with an unknown origin, genetics, epigenetics, and environmental triggers play an important role in this condition [117,118]. P2Y_12_ activation has been extensively investigated in animal models of RA. P2Y_12_-deficient mice have decreased osteoclast activity and they appeared to be protected from age-and tumor-associated bone loss [16]. Blocking the receptor with clopidogrel provides the same results in an animal model of RA [15,16], suggesting that the P2Y_12_ signaling is involved in bone metabolism. In this line, another study demonstrated that platelets can amplify the pathophysiology of RA by liberating proinflammatory microparticles that were detected in the human synovial fluid [119]. A recent clinical study reported that dual treatment of methotrexate, the most common drug used to treat RA, and ticagrelor improves the outcome of rheumatoid arthritis [120]. Either methotrexate or ticagrelor regulates adenosine extracellular concentration, and they appear to do so more efficiently when administered together. This study is very interesting, but it has certain limitations, such as a small sample size and a lack of controls. It is also noted that the majority of the patients were women [120]. In contrast with all the inflammation models previously reported, our group showed that, during erosive arthritis induced by peptidoglycan polysaccharide, the severity of the inflammation in the joints was augmented when animals were pre-treated with clopidogrel [15] or prasugrel [121]. This discrepancy between studies suggests that more investigations need to be carried out to understand how P2Y_12_ antagonism can alter RA and to determine whether P2Y_12_ antagonist exerts beneficial effects on this inflammatory condition.

### 6.3. Tumor Microenvironment

The tumor microenvironment (TME) is a complex of components, including immune cells, stromal cells, blood vessels, and extracellular matrices, and influences tumor growth, differentiation, and metastasis [122,123,124,125,126]. Immune cells are essential players during cancer-related inflammation. Modulating the inflammation in the TME has been suggested as a therapeutic strategy for cancer. Interestingly, cancer cells actively secrete ADP, thromboxane, and thrombin, which are all molecules that activate signaling pathways, including purinergic signaling [127]. This is probably one reason why there is a 20% risk of thrombosis in cancer patients. These observations lead to the question of whether P2Y_12_ is activated during cancer and, therefore, whether P2Y_12_ blocking can decrease the risk of tumor-associated thrombosis. Indeed, the recent TICON study investigating the effect of ticagrelor on platelets in vitro and in patients with metastatic breast or colorectal cancer concluded that ticagrelor could decrease platelet activation, decreasing the risk of thrombosis in cancer patients [128]. Next, it was investigated whether blocking P2Y_12_ could also modulate the interaction between the immune system and cancer cells [129]. For instance, ticagrelor has been investigated in both clinical studies and a murine model of breast cancer, showing that it could limit platelet-tumor interaction and metastasis [130].

Whether the effects noted by blocking P2Y_12_ are exclusively through platelets or other cells is still up for debate. In the TME, there is the presence of Tregs as well as macrophages [6] and both cell types have been shown to express this receptor [11], suggesting that blocking P2Y_12_ could alter the activation of multiple immune cells in the TME. It would be interesting to investigate whether blocking P2Y_12_ alters each one of the players in the TME and which mechanism is involved. This will provide useful information to address the question of whether P2Y_12_ blockers can be used as effective cancer treatment or co-treatment.

Overall, the data seem to agree that blocking P2Y_12_ alone or in dual therapy with aspirin could diminish cancer growth and metastasis. However, a clinical study reported an increase in cancer incidence and malignancy following dual treatment with prasugrel and aspirin [131]. As a result, more studies are required to investigate whether blocking the receptor P2Y_12_ could be beneficial during cancer.

### 6.4. Inflammation in Diabetes

Diabetes is a chronic disease inhibiting the body’s inability to produce insulin with vascular endpoints of stroke, nephropathy, and ischemic heart disease [132]. There is an emerging role of inflammation in the pathophysiology of diabetes [133]. Hyper-reactivity of platelets has been noted in diabetes mellitus which in turn contributes to complications of cardiovascular conditions [134]. Platelet hyperactivity in diabetes mellitus occurs with an increased expression of platelet receptors. Hu et al. (2017) reported enhanced expression of P2Y_12_ in rats and patients with diabetes mellitus, suggesting an increase in the activation of P2Y_12_ signaling [86]. As a result, P2Y_12_ antagonists with inverse agonist ability may have beneficial effects on the outcome of diabetes mellitus. Indeed, the P2Y_12_ antagonist ticlopidine was able to reduce macroaggregates in diabetic patients [135]. Chronic itching associated with diabetes in mice was reduced after blocking P2Y_12_ with ticagrelor or P2Y_12_ shRNA, indicating a change in the expression of P2Y_12_ through anti-inflammatory cytokine activities [136]. The inhibition of P2Y_12_ in combination with aspirin in patients with diabetes decreased ischemic events [137].

The mechanism has been investigated, and it has been reported that GPIIb/IIIa activation causes macroaggregates in diabetic patients due to the involvement of P2Y_12_. Interestingly, P2Y_12_ upregulation in diabetes has been linked to the activation of the ROS/NF-kB hyperglycemia pathway [86].

### 6.5. Pulmonary Inflammation and Asthma

Asthma is a chronic pulmonary inflammatory disease characterized by hyperresponsiveness of the airway to triggers such as allergens and viruses [138]. Dynamics in the biological response to allergens have made prediction and prevention quite challenging [139]. The interest in studying P2Y_12_ blockage in pulmonary inflammation and asthma started when it was discovered that platelets’ interaction with eosinophils and neutrophils in the lung tissue is a key step for pulmonary inflammation and lung injury [36]. Therefore, studies have been carried out to investigate whether blocking P2Y_12_ was beneficial for the outcome of pulmonary inflammation [13,140,141]. Several studies in animal models of asthma have indicated that blocking P2Y_12_ can decrease platelet-leukocyte interaction in the lungs, therefore diminishing cell infiltration [12,13,140,141]. This appeared to be beneficial for the outcome. Interestingly, the P2Y_12_ protein level was increased markedly in ovalbumin (OVA)-sensitized mice compared to the control [13]. This increase was not noted when OVA-sensitized mice were treated with clopidogrel [13]. Treatment with clopidogrel was overall able to reduce lung damage, alone or in dual therapy with other drugs [14]. However, experiments need to be performed on human subjects to verify the validity of this treatment. Platelets and eosinophils aggregate through the bound of P-selectin (on the platelet’s surface) and PSLG-1 (on the eosinophil cell’s surface) [142]. As the authors noted a decrease in platelet-eosinophil aggregation in the blood and the BAL fluid during asthma [13], it seems possible that blocking P2Y_12_ decreases P-selectin expression on platelets and as a result, reduces the chances of these cells to aggregate. However, more experiments need to be conducted to fully elucidate the mechanism or identify other pathways involved. The inflammatory response to LPS in mice was reduced upon the administration of ticagrelor consistent with patient samples [79]. The authors also demonstrated a greater effect of ticagrelor on survival and the anti-inflammatory response of ticagrelor over clopidogrel [143].

### 6.6. Sepsis

Sepsis is a complex clinical syndrome that occurs as a result of a serious infection associated with high morbidity and mortality [108]. The pathogenesis of sepsis is highly complex, and it involves multiple components of the immune system [108]. We have previously shown that the blockade of P2Y_12_ signaling in a mouse model of sepsis improves outcomes, most likely through decreased α-granule secretion of inflammatory mediators and reduced mobilization of P-selectin to the plasma membrane of platelets [12]. Further studies have shown that P2Y_12_ antagonism alters Treg population size and function in a stimulation-dependent manner [11]. The effects appeared to be through platelets but also by blocking P2Y_12_ in T lymphocytes. Previous studies have been performed on platelet depletion. For example, Asaduzzaman et al. (2009) investigated lung injury and neutrophil infiltration in sepsis when the mice had been platelet-depleted [102]. The data show that upon platelet depletion, the outcome of sepsis was improved. In a study of a myocardial infarction mouse model, the authors reported similar conclusions [94]. It would be interesting to repeat these experiments upon P2Y_12_ blockade in platelet-depleted mice. We [5,12] and others [83] have identified P2Y_12_-independent effects of clopidogrel and ticagrelor in immune cells, and these data should also be considered in analyzing the effect of P2Y_12_ blockers in inflammation.

Changes in P2Y_12_ levels during sepsis have been investigated by Zhong et al., (2021) [144]. They have shown that in the animal model of sepsis (cecal ligation and puncture, CLP) as well as in septic patients, P2Y_12_ protein levels are increased in platelets [144]. Interestingly, platelet reactivity appeared to be positively regulated to the P2Y_12_ content, suggesting that this could be the possible explanation for why platelets seem to be hyper-activated in sepsis. The samples from patients were obtained from both sexes, while the mice studied were male animals. It would be interesting to compare males and females to identify whether there is a sex-related difference. No change in other immune cells was evaluated. Similar data were observed in patients with diabetes mellitus, as we will be discussing below [86].

Not all studies agreed on showing that P2Y_12_ antagonism is effective in decreasing mortality and inflammatory levels in sepsis. Rabouel et al. (2021) did not see any amelioration in a mouse model of sepsis (CLP) upon P2Y_12_ blockage using clopidogrel or in mice with platelet-selective P2Y_12_ deficiency [104]. We have also reported that LPS-induced inflammation was more severe in P2Y_12_-deficient mice as compared with the control [103], suggesting that further experiments are required to investigate whether P2Y_12_ inhibition can be beneficial for sepsis.

### 6.7. COVID-19

COVID-19 is a respiratory disorder caused by the severe acute respiratory syndrome coronavirus 2 characterized by uncontrolled inflammatory responses leading to cytokine storm [145]. Thrombo-inflammation has been identified as a major cause of mortality and morbidity in patients with COVID-19 [146]. Platelets are hyper-activated in patients with COVID-19 [147] and this could explain the reason for the high risk of thrombosis. Interestingly, platelets from patients were more responsive to different agonists, including P2Y_12_ agonists [148]. Whether this is due to changes in P2Y_12_ expression on the platelet surface has not been investigated yet. However, studies have been carried out investigating whether antiplatelet aggregation therapy with P2Y_12_ blockers could help to prevent thrombotic events during COVID-19, and there they could also be re-purposed as a therapeutic strategy for COVID-19.

A recent study investigated the effects of antiplatelet aggregation therapy using different thienopyridine on the survival and outcome of COVID-19 patients. The authors show that targeting P2Y_12_ could decrease mortality and shorten the duration of mechanical ventilation [149]. However, another study showed that the P2Y_12_ blocker could not diminish the number of free-of-organ support days [150]. In both studies, the treatments did not seem to increase bleeding. Both sexes seemed also to be considered in the studies and no difference was noted. Interestingly, a more recent meta-analysis study did not discover any significant benefit from adding P2Y_12_ blockers to mortality and overall better outcomes. It was rather associated with major bleeding [151].

An increase in platelet interaction with other cells of the immune system, such as monocytes, neutrophils, and T lymphocytes, has been investigated in COVID-19 patients [148]. In severe cases of COVID-19, the patients showed an increase in platelet-neutrophil and platelet-monocyte aggregate formation [148]. However, the specific involvement of P2Y_12_ activation is still unclear.

Overall, the data suggest that more studies are required to determine whether blocking P2Y_12_ could be an additional tool for COVID-19 treatments. It would also be interesting to study whether this therapy can be effective in preventing post-infection complications and whether there are differences between thienopyridine.

### 6.8. Neuroinflammation

Neuroinflammation is a process related to various neurodegenerative disorders, such as Parkinson’s disease, Alzheimer’s disease, and multiple sclerosis [152]. Oxidative agents, trauma, redox iron, tau oligomers, and different viruses are some damaging signals which can trigger neuroinflammation. For example, the accumulation of amyloid plaques in Alzheimer’s disease has been viewed as a stimulus for microglia-driven neuroinflammation, but also as a consequence of immune senescence with microglial loss-of-function. Furthermore, it is characterized by hyperphosphorylation of the neuronal cytoskeleton protein tau, which leads to intraneuronal tau aggregates (so-called neurofibrillary tangles), and the pathological deposition of extracellular amyloid beta peptides (Aβ) (so-called amyloid plaques) from the amyloid precursor protein. Proinflammatory cytokines are secreted triggering hyperphosphorylation of the brain tau [153]. As P2Y_12_ can modulate microglia responses, blocking P2Y_12_ using clopidogrel has been investigated during neuroinflammation. Interestingly, following blood-brain barrier breakdown, microglial chemotaxis via P2Y_12_ induces the rapid closure of the blood-brain barrier by forming a dense aggregate at the site of the injury [17].

Studies have shown that P2Y_12_ signaling is activated in mouse models of multiple scleroses, such as EAE [18]. As mentioned in the T-lymphocyte section, P2Y_12_ activation appears to be correlated with Th17 differentiation. One study shows that receptor deficiency or blockage with either clopidogrel or ticagrelor ameliorated the EAE outcome [48]. Indeed, P2Y_12_ was upregulated in the spleen and lymph nodes, but not in the brain or the spinal cord at different time points [48]. The experiments were performed in male and female mice. However, another study has shown that EAE severity was increased in P2Y_12_ knockout mice [18].

## 7. Sex-Related Differences in P2Y_12_ Activation

Recent studies have investigated whether blocking P2Y_12_ provides different outcomes based on the sexes. First, it was studied whether targeting P2Y_12_ has the same effect in males and females in preventing further cardiovascular events in patients. Several studies have shown that the effects of ticagrelor or prasugrel in preventing cardiovascular events were comparable between men and women [154,155,156], and hence a sex-related adaptation of the therapy may not be required for cardiovascular disease treatment. However, other studies identified significant differences between male and female patients [157,158]. In particular, Ranucci et al. reported that upon treatment with clopidogrel, women’s platelet reactivity to ADP was higher than observed in men [157]. This appeared to be related to a higher platelet count, observed in women as compared to men. These data suggest that a higher dose of clopidogrel may be more indicated for women. Moreover, Waissi et al. analyzed blood samples from myocardial infarction male and female patients [159]. They measure platelet degranulation, fibrinogen binding, thrombin, and ADP-induced aggregation. Upon P2Y_12_ antagonism, female platelets appeared to be more reactive than male platelets. These data are very interesting, yet not conclusive, and hence more experiments need to be performed to clarify this discrepancy. Studies analyzing male and female patients side by side will be particularly meaningful.

There are studies in a variety of animal models investigating ticagrelor effects on inflammation [129,143,160,161,162], but most of the studies were performed on male mice [143,160,161,163]. Ticagrelor has been investigated in patients with inflammatory conditions, for example, in studies on inflammatory factors during myocardial infarction [164] or patients with pneumonia [143]. Both studies have shown that treatment with ticagrelor could decrease platelet activation, platelet-leukocyte interaction, and circulating cytokine levels in patients.

Several studies investigated the effects of sex hormones in platelet biology. As platelets express the receptors for the sex hormones [165,166,167], it seems plausible to believe these cells will be altered. Indeed, differences in platelet aggregation and functions have been noted between the sexes. The results are interesting yet still inconclusive [168]. One study investigated the effects of testosterone or estradiol in the expression of P2Y_12_ at the protein level in megakaryocytes [169]. The data show that exposure to testosterone, but not estrogen, increased P2Y_12_ receptor protein. Interestingly, the studies show comparable effects in males and, most likely, menopausal females [154,155,156]. These findings suggest that females may respond differently to P2Y_12_ blockers depending on estrogen levels. As a result, female age and estrogen levels should be considered in the studies, and medications prescribed accordingly.

## 8. Age-Related Differences in P2Y_12_ Activation

One can speculate that P2Y_12_ protein levels may change with age. Interestingly, the level of P2Y_12_ protein is comparable between platelets isolated from infants versus platelets isolated from adults. However, infant platelets seem to have enhanced P2Y_12_-mediated dense granule trafficking in response to ADP as compared to adult platelets [170]. Clopidogrel has been evaluated in infants and young adults at specific doses, and it was able to decrease platelet aggregation without causing serious bleeding [171]. In principle, this drug could be also effective in preventing cardiovascular diseases in infants and young adults. However, one should consider that CYP2C19 expression, which is needed to activate the prodrug, varies in the human ontology. Approximately 15% of mature levels throughout the prenatal period is found, and its expression increases linearly in the first five postnatal months. Adult CYP2C19 protein and activity values were observed in samples older than 10 years [102]. Age-related differences of other P2Y_12_ antagonists have not been studied so far.

P2Y_12_ protein content has not been investigated in the elderly. However, a number of studies analyzed whether P2Y_12_ blockers can be an effective therapy for the elderly at different ages. Overall, P2Y_12_ blockers such as clopidogrel [172] and ticagrelor [173]) were able to decrease the risk of cardiovascular diseases in the elderly but they both appear to have a high risk of bleeding [172,173]. Similar data were obtained for prasugrel, which has shown higher bleeding risks in individuals older than 65 [77]. As a result, despite being effective in decreasing cardiovascular events, these drugs may not be the optimal treatment for the elderly, and they need to be prescribed with caution.

## 9. Conclusions

In conclusion, P2Y_12_ activation appears to be a key step during inflammation, and hence blocking this receptor represents a promising therapeutic strategy that deserves further consideration (Figure 3). However, there are discrepancies between studies and, therefore, more studies need to be performed and most likely need to investigate each disease specifically (Figure 3). Whether the effects of blocking this receptor are exclusively through platelets or other immune cells is still up for debate, although a significant number of studies reported that immune cells were modified in their functions. Differences in P2Y_12_ activation and blockage between the sexes have been shown and they should be taken into consideration in all studies.

## Figures and Tables

**Figure 1 ijms-24-06709-f001:**
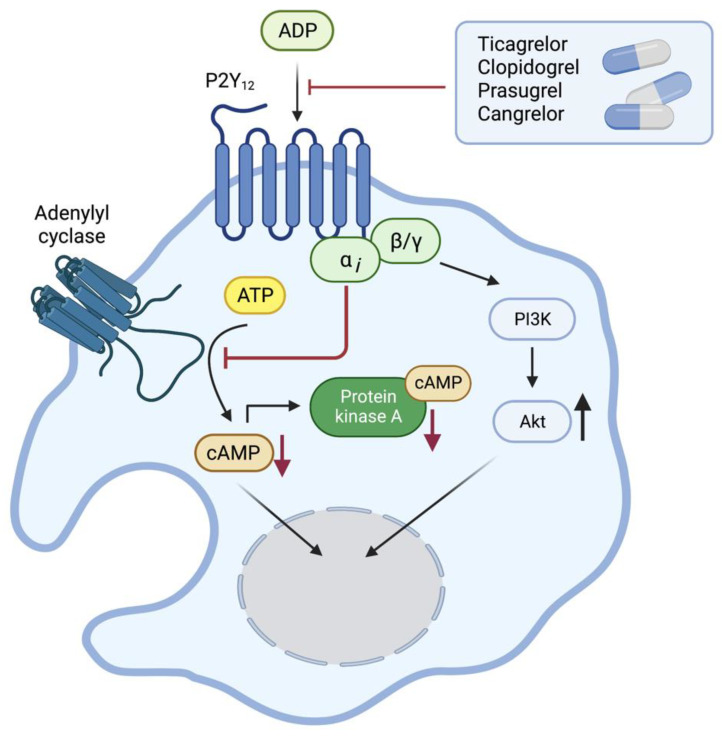
ADP-induced P2Y_12_ downstream signaling. ADP binds to and activates the ADP receptor P2Y_12_ leading to the inhibition of adenylate cyclases, a decrease in cAMP, PI3 kinase activation, and an increase in AKT phosphorylation. This signaling pathway eventually leads to platelet activation and aggregation. This figure has been generated using the software Biorender (https://www.biorender.com).

**Figure 2 ijms-24-06709-f002:**
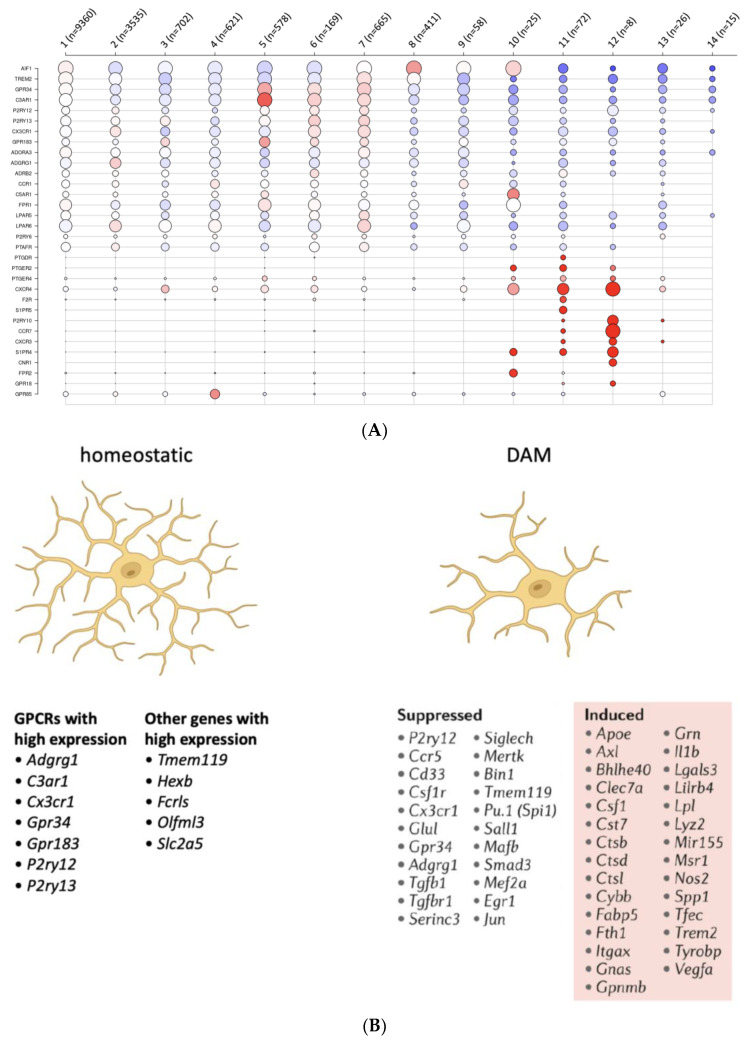
Expression of GPCRs and other selected genes in homeostatic microglia and disease-associated microglia (DAM). (**A**) Single-cell RNA sequencing and clustering of the mRNA expression pattern revealed 14 subpopulations of human microglia cells [58]. These subpopulations are equipped with different subsets of GPCRs and P2Y_12_ is found in all microglial subpopulations. Significantly expressed GPRCR mRNAs are shown (red = high, blue = low expression, and circle size reflexes the portion of the cells in the cluster expressing the gene). As references, the microglial markers IBA1 (AIF1) and TREM2 are given at the top of the list. (**B**) The homeostatic (resting) microglia express more than 30 GPCRs, and the receptors with the highest expression are listed. Together with additional genes (e.g., Tmem19, Hexb) these GPCRs are considered marker genes for homeostatic microglia. Transcriptome studies of microglia from patients and mouse models of neurodegenerative diseases revealed changes in the expression of some of these marker transcripts. This information is taken from different RNA sequencing studies [58,59,60,61].

**Figure 3 ijms-24-06709-f003:**
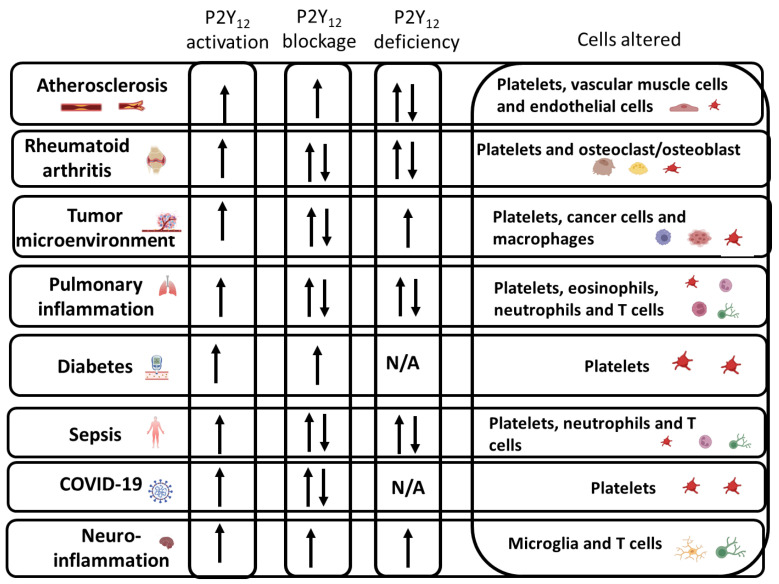
A schematic representation of how P2Y_12_ activation, blockage, and deficiency are altered in inflammatory conditions and a list of which cells appear to be affected. Overall, P2Y_12_ did appear to be activated in all the inflammatory conditions reported in the graph. However, it is still unclear whether blockage or deficiency could consistently improve the outcome. The arrow pointing top and down represents an increase and decrease respectively. Cells with both arrows shows inconsistency.

**Table 1 ijms-24-06709-t001:** P2Y_12_ activation in immune cells.

Immune Cell Type	P2Y_12_ mRNA	P2Y_12_ Protein	Functional Studies	Signaling	Discrepancy of Studies	Key References
**Platelets**	Detected	Detected	Platelet aggregation and secretion	AKT phosphorylation and decrease in cAMP	Consistency	[20,21,38,39,40]
**Monocytes**	Detected	Detected	Migration	Ca^2+^ mobilization	Not consistently detected in monocytes	[6,36,37,41,42,43,44,45,46]
**Macrophages**	Detected	Detected	Migration	Ca^2+^ mobilization	Consistency	[6,36,37,41,42,43,44,45,46]
**Neutrophils/eosinophils**	Not detected in neutrophils, no studies in eosinophils	Detected in eosinophils, not neutrophils	No studies	No studies	Consistency	[5,36,47]
**T lymphocytes**	Detected	Detected	Migration, differentiation (Th17 and Tregs), and cytokine secretion	No conclusive studies	Consistency	[7,11,18,48,49,50]
**Dendritic cells**	Detected	Detected	Endocytosis, Ag-presenting functions, IL-23 production	Ca^2+^ mobilization	Consistency	[4,18,49]
**Microglia**	Detected	Detected	Migration	PI3K, decrease in cAMP	Consistency	[41,42]
**Natural killer cells**	No studies	No studies	No studies	No studies	N/A	N/A
**B lymphocytes**	No studies	No studies	No studies	No studies	N/A	N/A

**Table 2 ijms-24-06709-t002:** Challenges and suggested solutions in studying the receptor P2Y_12_.

Challenges	Suggested Solutions
Reliable antibody	Control the data using different antibodies and against a KO mouse model.
Antagonist specificity	Select the concentration(s) more appropriate to avoid unspecific bindings; Confirm the data using two or more different antagonists.
Cell purity	Select a reliable method for cell isolations when working with primary cells; Verify the purity of your cell population with different methods.

**Table 3 ijms-24-06709-t003:** P2Y_12_ activation in the immune system during inflammatory conditions. This table provides a summary of what is known so far about the effects of P2Y_12_ blockage on a number of inflammatory diseases.

Inflammatory Disease	Animal Models	P2Y_12_ Antagonist	Outcome	Mechanism	Patients	Key References
**Atherosclerosis**		Clopidogrel Prasugrel Ticagrelor	Decrease plaque size and inflammatory molecules.	Blocking P2Y_12_ prevents platelet hyperactivation	Blocking P2Y_12_ appears to be beneficial.	[100,101,102,103,104]
**Rheumatoid arthritis**	Peptidoglycan polysaccharide (PG-PS)-induced arthritis model (rats)	Clopidogrel Ticagrelor P2Y_12_ KO mice	Not established	Unknown	Blocking P2Y_12_ appears to be beneficial.	[15,16,105,106,107]
**Tumor microenvironment**	In vitro Murine model of breast cancer Murine model of pancreatic cancer	Ticagrelor Prasugrel	Mostly decrease cancer growth and risk of thrombosis but not established	Decrease platelet activation and cancer growth	Blocking P2Y_12_ appears to be beneficial.	[6,9,86,87,88,89,90,91,92]
**Diabetes**	Rats Mice—high glucose and high-fat diet	Ticagrelor ticlopidine	Improvements in Cardiovascular diseases	Increase P2Y_12_	Blocking P2Y_12_ appears to be beneficial.	[68,103,104,105,106,108]
**Pulmonary inflammation and asthma**	OVA-induced asthma	Clopidogrel Ticagrelor	Decreased lung damage and diminished eosinophil infiltration.	Decrease cytokine secretion and eosinophil activation	Blocking P2Y_12_ appears to be beneficial.	[4,5,6,7,65,67,105,106,107,109,110,111,112,113,114,115,116]
**Sepsis**	Cecal ligation and double puncture (CLP) LPS-induced inflammation	Clopidogrel Ticagrelor P2Y_12_ KO mice	Not established	P-selectin surface expression T-cell differentiation	The effects of blocking P2Y_12_ are still unclear	[5,11,12,83,94,102,103,104,108]
**COVID-19**	N/A	Clopidogrel Prasugrel Ticagrelor	Not established	Unknown	Only patients	[86,89,91]
**Neuro-inflammation**	Experimental autoimmune encephalomyelitis (EAE)	Clopidogrel P2Y_12_ KO mice	Ameliorate EAE.	Th17 differentiation microglial chemotaxis	Only animals	[15,16,41]

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
