# Peer review of "The Signaling Pathway of the ADP Receptor P2Y12 in the Immune System: Recent Discoveries and New Challenges"

_ijms, 2023, doi:10.3390/ijms24076709_

Round 1

Reviewer 1 Report

The thrust of this article is that inhibitors of P2Y12 receptors may conceivably exert important anti-inflammatory effects outside the circulation. The basis for this suggestion is more or less limited to the fact that such receptors are present in a number of extravascular tissues.  I would not hugely mind this sort of speculation, except that the authors are either totally unaware of, or choose not to focus on, a number of aspects of the pharmacology of P2Y12 receptors and  antagonists acting on these receptors. Specifically:-

(1) agents such as clopidogrel and ticagrelor act primarily by modulating adenylate cyclase activity. This is not mentioned in the text, and yet it is known that ADP binding to P2Y12 receptors inhibits adenylate cyclase signalling, both indirect (for example by prostacyclin and adenosine) and direct (for example by forskolin). Indeed, potency of agents such as clopidogrel and ticagrelor is directly proportional to pretreatment platelet anti-aggregatory response to anti-aggregatory prostanoids such as PGE1 and prostacyclin. That said, the potential utility of P2Y12 receptor antagonists outside platelets is also very likely to depend on extent of prostacyclin effects. I cannot understand why this central issue was omitted.

(2) The post-receptor G-protein- mediated effects of ADP, and therefore of P2Y12 antagonists, include biased signalling via Gi proteins, at least within platelets. Has it been determined whether this applies outside platelets? Obviously a question of some importance.

(3) Utilization of P2Y12 receptor antagonists is constrained because of the risk of bleeding, especially with more aggressive treatment regimens. This is resulting, for example, in the withdrawal of prasugrel from the market. I think that concentration: response characteristics for all of the candidate actions of candidate antagonists beyond platelets need to be provided.

There are also minor but significant spelling errors. For example, line 140 should read "though" not "through".

Reviewer 2 Report

Entsie et al’s paper shows the importance of the P2Y12 and the P2Y12 receptor drugs in different diseases. They emphasise the clinical reference of the receptors and their presence on the immune cells. Also have a short look at the problems of the investigation of this receptor types. However, the information is not well structured in the manuscript. I missed the logical line sometimes. A reread and a new order of the segments may be useful to make it more readable.

Questions and comments:

1)  line 49: Nor cAMP neither PI3 abbreviations are not solved

2) line 62: vWF is not solved

3) Figure 1.: Akt is not mentioned in the figure legend- All in all the legend could be more detailed or at least could mention the illustrated processes.

4) Table 1.: the rows could be ordered in some point of view (I.e. innate or adaptive immun cell type or origin or at least following the next paragraphs titles)

5) lines 86-88: The cited paper [9] also mentions it is a crosstalk between the P2Y1 and P2Y12. It should be mentioned here, too, as previously the authors said P2Y12 connected to Gi proteins and there is no information how Ca2+ could be influenced in this case.

6) line 89: The key words for database search should be written here.

7) line 92: Abbreviation TAM is unnecessary, as it is not used later.

8) lines 101-103: “Taken together, these data suggest that P2Y12 mRNA can be found in both monocytes and macrophages, but it was detected more consistently in macrophages.” mRNA was not investigated (or mentioned by the authors) in case of macrophages. They said it is expressed – but it could be proven by showing the protein of the P2Y12. It needs to be clarified.

9) line 109: Th17 cells should be solved.

10) line 129: WT is not solved (but solved some lines below).

11) line 130: ADP induced Ca2+ mobilisation is here again.

12) line 143: Starting the sentence with small letter instead of capital.

13) line 162: (DAM) is unnecessary.

14) chapter 3.5: The authors mention age and sex dependence, but do not go in details (but these are interesting topics)

15) chapter 3.5: Here the authors start to use the GPCR abbreviation, which was not introduced and the previous sections simply G protein receptors were used.

16) figure 2: Proteins and gene names are used alternatively which is disturbing (on the left side). The right side of the figure is not readable.

17) line 189: How “ADP-induced contractions” happens normally? (Also, how the migrations happen?)

18) line 197: Why the splenic artery is important? What happen if it is a functional receptor? What the authors think about it?

19) line 227: “Indeed, it may not be the best drug for the elderly.” - please, go in details.

20) line 234-236: please, go in details.

21) table 2: There is no reference for the rheumatoid arthritis and too much abbreviations were used, which is not solved. There is a place for a longer table legend.

22) line 316: What is methotrexate? – add some information about it, please.

Questions:

Which enzymes are involved in the activation of prasugrel?

Why do the authors think cell lines are important in research? If a drug get inside the body there not only one-type of cell will be, in this point of view “whole body models” are preferred than cell lines.

Next to sexes what other componenst would affect the expression of the P2Y12? Age would not be more important? As far as I know the cardiovascular diseases catch the women later because the oestrogens have defensive effect. Could it be that the women group were older than the men group? How were the age groups published in the cited studies?

Other opinions/notes:

1)     Some parts other letter-style shows up (e.g. line 49: Fig.1 or line 50: short-term).

2)     The diseases, syndromes could be introduced in some words/sentences in the section 5.

3)     Section 5 could be ordered in some point of view (i.e.: acute/chronic or other).

4)     More illustrations would make the paper nicer.

5)     I feel the paper poorly organised. It starts with the receptor and its expression – maybe I put after it how difficult it is to investigate it, and after taking a step to the clinical part.

Reviewer 3 Report

The manuscript is a compreensive review of what is known about P2Y12 expression in immune cells  and the effect of P2Y12 activation and/or inhibition in inflammatory conditions.  Authors should revise the text due to the existence of some discrepancies in font size.

Round 2
